# Talkin’ Toxins: From Coley’s to Modern Cancer Immunotherapy

**DOI:** 10.3390/toxins12040241

**Published:** 2020-04-09

**Authors:** Robert D. Carlson, John C. Flickinger, Adam E. Snook

**Affiliations:** Department of Pharmacology and Experimental Therapeutics, Thomas Jefferson University, 1020 Locust Street, Philadelphia, PA 19107, USA; Robert.Carlson@jefferson.edu (R.D.C.); John.Flickinger@jefferson.edu (J.C.F.J.)

**Keywords:** cancer, immunotherapy, vaccine, immune checkpoint inhibitors, adoptive cell therapy, cytokine therapy, Coley’s Toxins

## Abstract

The ability of the immune system to precisely target and eliminate aberrant or infected cells has long been studied in the field of infectious diseases. Attempts to define and exploit these potent immunological processes in the fight against cancer has been a longstanding effort dating back over 100 years to when Dr. William Coley purposefully infected cancer patients with a cocktail of heat-killed bacteria to stimulate anti-cancer immune processes. Although the field of cancer immunotherapy has been dotted with skepticism at times, the success of immune checkpoint inhibitors and recent FDA approvals of autologous cell therapies have pivoted immunotherapy to center stage as one of the most promising strategies to treat cancer. This review aims to summarize historic milestones throughout the field of cancer immunotherapy as well as highlight current and promising immunotherapies in development.

## 1. Introduction

The understanding of immune system governance in neoplastic growth and development has made significant leaps in recent years [1], but its origins can be traced back well over a century ago. Incidence of tumors spontaneously regressing following infectious or pyretic periods have been described throughout history [2,3,4]. However, advancements made in histological diagnosis and assessment of tumor malignancies over the past 100 years have given credence to these claims of immune system modulation in cancer.

## 2. Pivotal Observations in Cancer Immunotherapy

It is possible that cancer has existed ever since the evolution from unicellular organisms into multicellular entities. However, the oldest record of cancer to date is from a 240 million-year-old fossil containing a shell-less stem turtle, *Pappochelys rosinae*, with evidence of osteosarcoma [5]. Until recently, the treatment of cancer has historically focused on tumor excision, cytotoxic chemotherapeutic agents, and radiation therapy. Only after the turn of the 21st century did immunotherapy to treat cancer take stage [Figure 1].

### 2.1. The Story of Coley’s Toxins

William Coley, often regarded as the “Father of Immunotherapy”, was a bone surgeon in New York from 1890–1936 who famously developed a cocktail of heat-killed bacteria, called “Coley’s Toxins”, to treat patients with osteosarcoma. Inspiration for developing this treatment apparently started with one of his first patients, a young woman with osteosarcoma of the hand. Despite his surgical intervention (amputation of the forearm), she succumbed to metastatic disease within months of the operation. This episode had a profound impact on Coley and motivated him to learn more about her disease. He began by reviewing hospital medical records from ninety sarcoma patients, an analysis he later published [6]. While conducting his review, one patient’s course of disease was of particular intrigue. Coley came across the description of a patient with an inoperable sarcoma whose tumor completely regressed after developing erysipelas [7], a type of skin infection [8]. Upon reading this account, Coley wondered if it was possible to induce erysipelas in patients as a means to treat cancer. Fortunately for Coley, a German surgeon named Friedrich Fehleisen had only a few years earlier, in 1883, identified *Streptococcus pyogenes* as the bacterium responsible for erysipelas [9]. Thus, Coley was able to test his hypothesis and began injecting sarcoma patients with *Streptococcus pyogenes,* a primitive version of what would later be named Coley’s Toxins.

Over the course of Coley’s career, from 1888–1933, he tested over a dozen different preparations of his toxin. Developing his infamous toxin required striking a balance between safety and efficacy. Indeed, early preparations were highly variable. Some preparations were impotent and failed to produce any signs of infection while other preparations were highly infectious and led to mortality [10]. Eventually, Coley settled on a combination of heat-killed *Streptococcus pyogenes* and *Serratia marcescens* [11]. Although Coley was not the first person to make a connection between infection and cancer regression, nor the first to inject bacteria into a patient as a means to mediate tumor rejection, Coley’s efforts were the most comprehensive and influential. In total, it is estimated that Coley himself injected more than 1000 cancer patients and published over 150 papers related to the topic [11].

Coley reported remarkable success with his toxins and published many reports of his toxins inducing tumor regression [12,13]. However, at the time, his findings were highly controversial and were met with harsh criticism by many of his colleagues. Notable critiques include those in the *Journal of the American Medical Association* in 1894 issuing a statement criticizing the use of his toxins as well as the FDA re-categorizing of “Coley’s Toxins” in 1963 as an investigational drug that lacked safety and efficacy data, despite over 70 years of use and numerous publications [11]. This recategorization made it illegal to prescribe Coley’s Toxins outside of clinical trial testing. In the end, history would be on the side of William Coley. Years after his death, his toxins were re-evaluated in a controlled trial and were demonstrated to mediate antitumor effects [14]. Moreover, advancements in fundamental understanding of cancer and the immune system have allowed his findings to become more widely accepted and to lay a foundation for future studies of cancer immunotherapy.

### 2.2. Evidence the Immune System Targets Cancer

Although Coley never fully understood the mechanism by which his toxins functioned, he gathered substantial evidence linking the immune system and cancer. Further clarity and development of this connection would come years later in the form of the immunosurveillance hypothesis. The idea that the immune system possesses a capacity to recognize and eliminate cancer cells was first postulated by Paul Ehrlich in 1909 [15]. While direct experimental evidence during this time period was lacking, Ehrlich reasoned that the incidence of cancer is relatively low but that the formation of aberrant cells is a common phenomenon, suggesting the existence of a host defense system against cancer. Over 50 years later, these ideas were further developed by Burnet and Thomas and formally coined the “immune surveillance” hypothesis [16,17].

Early experimental evidence for the existence of tumor-specific immunity derives from transplantation studies. In 1943, Luwik Gross utilized methylcholanthrene (MCA) to chemically induce sarcoma in a C3H mouse and then transplanted this sarcoma into syngeneic mice. While inoculation with high doses of tumor cells often killed mice, Gross found that inoculation with low doses of tumor cells led to a period of growth followed by gradual tumor regression. In these surviving mice, tumor challenge using high doses of tumor cells invariably led to rejection, suggesting these animals developed immunity to the tumor [18]. Further support for immunosurveillance comes from a seminal study by Prehn and Main in 1953. In these studies, an array of sarcomas from multiple syngeneic mice were generated using MCA. Prehn and Main found that inoculation of a mouse with sarcoma from one source protected that mouse from future challenge using the same sarcoma source but did not protect against challenge using sarcoma derived from a different mouse [19]. Moreover, Prehn and Main demonstrated that transplantation of skin tissue from a donor mouse did not sensitize the recipient mouse to the donor’s sarcoma, directly addressing a common critique at the time that rejection was mediated by subtle differences in genetic backgrounds. Collectively, these studies further supported the existence of tumor-specific immunity, adding the nuance that tumor antigens are highly unique to a tumor even in tumors of the same histological type, induced by the same chemical means, and from mice of the same genetic background [19].

While studies in partially immunocompromised mouse models over the following decades failed to support the immunosurveillance hypothesis, definitive demonstration of immunosurveillance came in the early 2000s following a series of studies conducted in novel, specifically immunocompromised, mouse strains. In 2001, Robert Schreiber’s group compared the incidence of spontaneous neoplasms between wild-type and *Rag2*^−/−^ mice (*Rag2* encodes a protein necessary for somatic recombination and thus *Rag2*^−/−^ mice lack mature T and B lymphocytes) [20]. In mice over 15 months old, fewer than 20% of wild-type mice contained neoplastic disease while 100% of surveyed *Rag2*^−/−^ mice developed spontaneous neoplastic lesions in various tissues, suggesting functional T and B lymphocytes suppress the development of cancer. Moreover, the same study observed that *Rag2*^−/−^ mice, as well as *Ifngr1*^−/−^ and *Stat1*^−/−^ mice, which are deficient in vital immune signaling pathways, develop higher incidences of sarcoma compared to wild-type mice in MCA-induced tumor models [21]. Similarly, higher incidences of MCA-induced tumors were reported by additional investigators using mice deficient in other vital immune-signaling molecules such as perforin or TNF-related apoptosis-inducing ligand (TRAIL) [22,23]. These experimental studies in mice are mirrored by clinical evidence that humans with compromised immune systems develop higher incidences of cancer. Indeed, individuals born with genetic defects in immune-related genes develop higher incidences of lymphoma [24]. Moreover, people with otherwise normal immune function who acquire AIDS infection or transplant patients who receive immunosuppressive drugs are both at higher risk for developing Non-Hodgkin’s lymphoma and virus-induced Kaposi Sarcoma [25,26].

Since the emergence of the immunosurveillance hypothesis, the interplay between the immune system and cancer has been further refined and renamed as the process of “immunoediting” [27]. Immunoediting posits that the immune system and cancer intersect at three stages: elimination, equilibrium, and escape. During the elimination stage, the immune system recognizes and destroys many, but not all aberrant cells. During equilibrium, the immune system and the tumor exert opposing forces, effectively resulting in containment of the tumor. Over time, as the cancer acquires additional mutations and as the immune system exerts a selective pressure eliminating immunogenic cells and leaving behind non-immunogenic cells, the cancer eventually fully escapes immune surveillance. In this final escape stage, the cancer has fully circumvented detection by the immune system and undergoes rapid and uncontrolled growth. Evidence for equilibrium and escape stages is supported by experiments in mice with stagnant tumor sizes who then undergo rapid growth after immune cell depletion [28]. Additionally, tumors arising from immunocompromised mice are more frequently rejected when transplanted into a wild-type host compared to tumors arising from immunocompetent mice [21], partly reflecting immune-induced antigen loss in the presence of an intact immune system [29] (immunoediting). Thus, the fundamental goal of cancer immunotherapy is to overcome the years to decades of immunoediting to generate antitumor immunity that is sufficient to completely eliminate the patient’s cancer and cure their disease.

## 3. Immunomodulatory Agents

Immunomodulators comprise a variety of therapeutic agents and treatment strategies that aid in normalizing, or in the context of cancer, re-engaging or boosting immune cell function to counter uncontrolled cell proliferation. By definition, a tumor that presents in the clinic can be said to have escaped normal immune control, even if tumor-reactive T cells are detected in the blood or have infiltrated the tumor tissue [30]. Although tumor cells themselves possess intrinsic immunosuppressive behaviors, such as cultivating a hypoxic microenvironment [31,32] or generating lactic acid [33], the majority of suppressive influence comes from the normal functions of suppressive immune cells, cytokines, and inhibitory surface molecules [34,35]. Under normal physiological circumstances, these mechanisms suppress T-cell priming and cytotoxic T-cell function to stave off unwarranted autoinflammatory responses [36]. Therefore, many of the immunomodulatory agents described herein aim to directly oppose these immunosuppressive mechanisms and to re-engage the immune system.

### 3.1. Cytokine Therapy

Prior to understanding their therapeutic immunomodulatory potential in cancer, cytokines were first recognized as systemic soluble factors that could regulate lymphocyte function and inflammatory responses. In 1972, a group from Yale University School of Medicine first characterized a “lymphocyte activating factor” that spurred lymphocyte proliferation in response to soluble agents released by other syngeneic immune cells [37]. These agents were partially purified from antigen-stimulated, lymphocyte-conditioned media and characterized further as “T-cell growth factor” that could support cytotoxic T cells capable of killing autologous leukemic myeloblasts [38]. Shortly thereafter, this key growth factor was definitively purified and described as interleukin-2 (IL-2) [39], which not only allowed T lymphocytes to be cultured in large quantities ex vivo, but also allowed recombinant IL-2 to be administered as a high-dose single-agent [40], or used in tandem with preconditioned and transplanted cancer-specific lymphoid cells [41,42]. A comprehensive report published in 1987 by the NCI’s Surgery Branch documented objective responses to high-dose IL-2 and regression of tumors in patients with metastatic melanoma and renal cell cancer [43]. Adjustments made to IL-2 dose scheduling would largely combat acute toxicities, the most prominent being capillary leak syndrome and hypovolemia [44]. These results and safety measures would spur numerous and larger cohort studies utilizing IL-2 in a metastatic setting [45,46], culminating with FDA approval of high-dose intravenous IL-2 for patients with metastatic renal cancer in 1992, and metastatic melanoma several years later. This would be noted as one of the first FDA-approved cancer immunotherapies [47].

Other cytokines that have demonstrated translational applications include interleukin-15 (IL-15), interferon-alpha (IFNα), and granulocyte macrophage colony-stimulating factor (GM-CSF). While both structurally similar and capable of stimulating early T-cell proliferation and NK cell engagement much like IL-2, IL-15 additionally supports memory CD8^+^ T-cell persistence, a known mediator of long-term antitumor immunity [48]. IL-15 has also proven to be an effective mediator of antitumor protection in murine models of cancer [49,50,51,52]. In support of these claims, phase 1 clinical trials utilizing recombinant IL-15 alone, and in conjunction with B-cell-depleting antibodies, are currently underway for treating both solid and liquid tumors, respectively [NCT01021059, NCT03759184].

IFNα, another cytokine originally described in the context of mediating antiviral immune response [53,54], was also identified to inhibit tumor neovasculature, upregulate MHC class I expression, mediate dendritic cell maturation, activate B and T cells, and induce apoptosis—all favorable antitumor attributes [55]. Thanks to the efforts of blood banking that began in the late 1970s [56], adequate quantities of purified IFNα spurred a burst of clinical evaluations in patients with hematological malignancies [57,58] and solid tumors, such as renal cell cancers and malignant melanoma [59,60]. These trials culminated in FDA approval of IFNα as an adjuvant therapy first in rare forms of leukemia, and later in patients with high-risk stage II and stage III melanoma [61].

As the name suggests, GM-CSF was originally identified as a regulator of granulocyte and macrophage differentiation, as well as general hematopoiesis of multi-lineage progenitors [62]. However, in 1993, Dranoff and colleagues transduced B16 melanoma cells with ten known pro-inflammatory mediators, vaccinated mice with these constructs, and then challenged them with live B16 cells. Of the ten, GM-CSF conferred the largest magnitude of antitumor immunity [63]. These findings prompted strategies to deliver GM-CSF to patients, either by vaccinating patients with irradiated tumor cells engineered to secrete the cytokine [64], or by single-agent dosing [65]. Although the exploration of GM-CSF-expressing tumor vaccines has waned in recent years due to limited clinical efficacy [66], combination strategies employing recombinant GM-CSF with other immunomodulatory agents, such as checkpoint inhibitors [67] and additional cytokines [68] have enhanced overall survival in melanoma patient trials.

### 3.2. Immune Checkpoint Inhibitors

Suppression of immune cell activation, infiltration, and effector functions required for tumor cell clearance can be largely attributed to the immunosuppressive conditions within the tumor microenvironment [69]. In several studies, “dysfunctional” CD8^+^ T cells were retrieved from patient tumors and nearby draining lymph nodes and said to be lacking the expected differentiation profiles [70], or impaired by the accumulation of repressive Foxp3^+^ regulatory T cells (Tregs) [71]. A similar phenomenon was originally described in mice infected with a lymphocytic choriomeningitis virus, whereby chronic antigen stimulation induced an “exhausted” T-cell state proceeded by T-cell receptor (TCR) downregulation [72]. The discovery of other requisite activating co-stimulatory signals in addition to canonical antigen stimulation with the TCR [73], which was now known to be insufficient for fully functional T-cell activation [74], gave clues to the complex nature of balancing activation with self-antigen tolerance [75]. Immune homeostasis is dependent upon these co-stimulatory/co-inhibitory receptor-ligand interactions, which in the correct context, safeguard against chronic immune activation and excessive inflammatory responses [76]. Addressing these phenomena directly, immune checkpoint blockade selectively restricts these co-inhibitory signaling mechanisms that have been co-opted by cancer cells, thereby enhancing antitumor T-cell activity.

Although initially identified in 1987 [77], the 1994 discovery of cytotoxic T lymphocyte-associated antigen-4 (CTLA-4) co-inhibitory receptor pairing with the B7 co-stimulatory ligand is perhaps the most substantial [78]. Upregulation of CTLA-4 on both CD4^+^ and CD8^+^ T lymphocytes was identified as a negative regulator of T-cell activation and effector functions [79,80], while murine models deficient in CTLA-4 experienced massive lymphoproliferation and tissue infiltration due to over-activation of resident T cells [81,82]. In the late 1990s, Dr. James Allison’s group at University of California, administered an inhibitory antibody to block the CTLA-4 co-inhibitory synapse in mice burdened with tumors. Both orthotopic and pre-established tumor cells were rejected following administration of the anti-CTLA-4 antibody, indicating that blockade of inhibitory signals associated with the co-stimulatory pathway can enhance antitumor immunity [83]. These indications prompted the application of CTLA-4 blockade in patients with stage III/IV unresectable melanoma with remarkable success [84], culminating in the 2011 FDA approval of the anti-CTLA-4 monoclonal antibody (mAb), ipilimumab, as an adjuvant therapy for patients with cutaneous melanoma. Retrospective studies have revealed marked increases in survival benefit compared to traditional chemotherapy regimens [85,86], with modest gains observed in other solid tumor types currently in various phases of clinical trials [87].

In an effort to identify genes associated with apoptosis, Dr. Tasuku Honjo’s group at Kyoto University discovered programmed cell death protein 1 (PD-1), a lymphoid cell surface protein that the group hypothesized to be a cell-death inducer [88]. Several years later in 1999, the same group generated a PD-1 deficient mouse model that spontaneously developed several autoimmune-like symptoms and systemic graft-versus-host-like disease [89]. Like CTLA-4, PD-1 was identified as a negative regulator of adaptive immune responses. PD-1 ligand 1 (PD-L1) was discovered that same year at the Mayo Clinic and characterized as functionally homologous to the CTLA-4 ligand, B7, but co-stimulated T cells through some additional unknown receptor [90], later identified to be PD-1 [91]. Engagement of PD-1 with its ligand prevented T-cell proliferation and cytokine production when synthetically stimulated, identifying it as an intrinsic inhibitory mechanism of autoreactive lymphocyte activation [91]. PD-L1 surface expression on tumor cells was also discovered to suppress the cytolytic effector functions of CD8^+^ T cells, with additional speculation that PD-1/PD-L1 blockade could serve as an effective strategy to combat tumor cell escape [92]. Speculation became reality when several groups tested PD-L1 blockade in murine tumor models and concluded that antibodies directed at this co-stimulatory interaction could enhance cancer immunotherapy [93,94,95]. In one step closer to the clinic, PD-L1 was determined to be a prognostic marker of patient outcome, with higher levels of ligand in resected specimens correlated with poorer patient survival [96]. Within the past 10 years, several high-profile trials employing anti-PD-1/PD-L1 mAbs under various conditions, dosing strategies, and cancer types, have indicated that blockade of this co-inhibitory pathway is both well-tolerated and associated with durable objective responses in patients [97,98,99]. Consequently, FDA approval was granted first to nivolumab, a humanized PD-1 mAb for metastatic melanoma in 2014, and subsequently for pembrolizumab, a PD-1 mAb alternative. Both therapies elicited greater overall patient survival compared to their anti-CTLA-4 counterpart [100,101]. In 2016, a third antibody was developed, this time directed at the PD-L1 ligand to treat patients with urothelial carcinoma and non-small cell lung cancer with much success [102,103]. This fully humanized anti-PD-L1 mAb, atezolizumab, was granted FDA approval for bladder cancer patients ineligible for traditional cisplatin-based chemotherapies [104]. Recent studies have expanded the number of indications for anti-PD-1/PD-L1 blockade alone [105], and in combination with anti-CTLA-4 [106], both proving to be summarily efficacious.

Although CTLA-4 and PD-1 blockade strategies have demonstrated unprecedented clinical success and accelerated FDA approval, there remains a population of non-responders. These individuals either fail to respond to checkpoint blockade from treatment onset due to innate resistance mechanisms, or acquire secondary resistance resulting in relapse. Retrospective studies of large-cohort clinical trials may expose novel biomarkers capable of predicting resistance to checkpoint therapies [107]. Additional co-inhibitory receptors, each with unique functions, have since been identified to influence negative immune regulation by various mechanisms [108]. Likewise, recent findings have demonstrated that the resident gut microbiome has the ability to influence patient responses to checkpoint blockade, with individuals who had consumed oral antibiotics prior to therapy experiencing poorer anti-PD-1 responses [109,110]. Nonetheless, immune checkpoint inhibitors continue to represent the vast majority of new immunotherapies for the treatment of cancer. These therapies would not be possible without seminal discoveries made in the blockade of negative immune regulatory elements by Drs. James Allison and Tasuku Honjo, for which they were awarded the 2018 Nobel Prize in Physiology or Medicine.

## 4. Vaccines

While immunomodulatory agents broadly stimulate the immune system, cancer vaccines aim to more precisely steer an immune response towards cancer. At its purest form, a cancer vaccine consists of one or more tumor antigens combined with an adjuvant to enhance the immune response. As will later be described, the type of tumor antigen, delivery method of the antigen, and adjuvant varies greatly. Cancer vaccines can be administered as a therapeutic vaccine in patients with existing malignancies or as a preventive vaccine in healthy or high-risk individuals (primary prevention) or patients in remission (secondary prevention) to protect against future tumor development or recurrence, respectively.

### 4.1. Tumor Antigens

Initial attempts at vaccination occurred before the identification of specific tumor antigens. These trials utilized cellular-based vaccines consisting of modified or irradiated tumor cells derived from a patient (autologous) or from a cancer cell line (allogeneic) injected with adjuvant [111,112,113]. With the identification of the first human tumor antigen, MAGE-1, by Thierry Boon’s group in 1991, a more refined approach of vaccinating against specific targets was born [114]. Since then, over 75 tumor-associated antigens have been identified [115]. There are two categories of tumor antigens: tumor-associated antigens and tumor-specific antigens. By definition, tumor-associated antigens share expression with some normal tissues while tumor-specific antigens are unique to cancer cells. Notable examples of tumor-associated antigens that have been a focus of multiple immunotherapies include the cancer-testis antigens NY-ESO-1 [116] and MAGE-1 [117], which are expressed by germ cells and ectopically re-expressed in cancers; the oncofetal antigens CEA [118] and alpha-1-fetoprotein [119], which are present during fetal development and re-expressed by cancers; differentiation antigens, such as prostate-specific antigen (PSA) [120] and CD19 [121], which are expressed by cells derived from a specific tissue-type and retained in cancers; and antigens that are over-expressed by cancers relative to normal tissue, including HER2 [122] and telomerase [123]. In contrast to tumor-associated antigens, which share expression with healthy tissue, tumor-specific antigens are exclusively expressed by tumors. Tumor-specific antigens, also known as neoantigens, are mutated peptides created by unique genetic aberrations or may be viral antigens in the case of virus-associated cancers [124].

### 4.2. Therapeutic Cancer Vaccines

In 1995 and 1996, the first clinical trials testing cancer vaccines against tumor-associated antigens were published. These trials utilized either peptide vaccines [125], or peptide-pulsed dendritic-cell vaccines composed of patient-derived dendritic cells that have been incubated with a peptide prior to re-infusion [126,127]. Other popular methods of cancer vaccination include the use of recombinant viral and bacterial vectors. As microorganisms potently stimulate the immune system, the use of these vectors to deliver tumor antigen in the context of an infection is hypothesized to enhance antitumor immune responses. Common vectors used to deliver tumor antigens include poxvirus [128], adenovirus [129], *Salmonella typhimurium* [130], and *Listeria monocytogenes* [131]. As methods of gene therapy have advanced over the years, the use of DNA and RNA vaccines has become increasingly common [132].

Despite thousands of cancer vaccine clinical trials, only one therapeutic cancer vaccine, Sipuleucel-T, is FDA-approved [133]. Sipuleucel-T is an autologous cell vaccine composed of patient peripheral blood mononuclear cells (PBMCs) pulsed with a chimeric protein consisting of the tumor-associated antigen prostate alkaline phosphatase (PAP) fused to the immunomodulating cytokine GM-CSF. A phase III clinical trial in men with metastatic castration-resistant prostate cancer found that three infusions of Sipuleucel-T led to the induction of PAP-specific immune responses and a 4.1-month improvement in median survival [134].

Limited success in therapeutic settings may be, in part, attributed to poor immunogenicity of the vaccine target and immunosuppressive tumor microenvironments. One approach to overcome poorly immunogenic tumor-associated antigens is the recent trend towards targeting neoantigens. Compared to tumor-associated antigens, neoantigens may be more immunogenic due to a lack of immunological tolerance mechanisms [135]. Until recently, the identification of neoantigens was impractical as the cost and time to sequence patient genomes for unique mutations presented a formidable barrier. However, with advancements in next-generation sequencing, it has become feasible to sequence a patient’s normal and tumor genome to identify unique tumor-specific antigens. Personalized therapeutic neoantigen vaccines have shown promise in phase I trials for melanoma [136] and glioblastoma [137]. However, these neoantigen vaccines are in early clinical testing, and thus the efficacy and feasibility of this approach is yet to be determined.

### 4.3. Preventive Cancer Vaccines

Recently, there has been a trend towards testing cancer vaccines as preventive therapies. Vaccination in preventive settings may be preferable to therapeutic ones as it may allow for the induction of antitumor immunity before the development of immunosuppressive microenvironments [138]. This strategy has shown promise against multiple viral-based cancers. Indeed, vaccination against oncogenic viruses including hepatitis B and human papillomavirus have led to reductions in hepatic [139] and cervical [140] cancers, respectively. However, for non-virally associated cancers, a target antigen and clinical setting to administer preventive vaccines is often less clear. For example, vaccinating a healthy patient against a tumor-associated antigen may carry an unnecessary risk of autoimmunity. Additionally, preventive vaccination against neoantigens, while reducing the risk of autoimmunity, may be impractical as neoantigens are often widely variable between patients. However, preventive vaccination in some settings may be possible. One such example is vaccination against the mucin 1 (MUC1) antigen in patients at high-risk of colorectal cancer. In tumors, MUC1 is hypoglycosylated relative to normal tissues, allowing for the induction of selective antitumor responses [141]. A phase I/II study in patients with a history of colorectal adenoma demonstrated MUC1 immunogenicity and a phase II trial investigating the ability of MUC1 vaccine to prevent adenoma recurrence is currently ongoing [142].

### 4.4. Oncolytic Virotherapy

An emerging immunotherapeutic strategy that is often categorized as a cancer vaccine is the use of oncolytic viruses. Oncolytic viruses preferentially infect and kill tumor cells compared to normal tissue. Selective infection of tumor cells is achieved through a combination of factors including the overexpression of viral receptors on tumor cells which can facilitate viral entry, a proliferative cell cycle that promotes viral replication, and a tumor cell deficiency in type I interferon production leading to limited viral clearance [143]. In addition to mediating tumor regression by direct cell lysis, viral infection activates components of the innate and adaptive immune system, thereby contributing further to tumor cell death. For example, oncolytic viral infection activates NK cells to clear virally-infected tumor cells [144]. Moreover, immunogenic cell death of virally infected tumor cells releases both tumor-associated antigens and neoantigens that can be acquired and presented by antigen-presenting cells, leading to the induction of antitumor CD8^+^ T cell responses (an approach often described as “in situ vaccination”) [145,146].

The potential of oncolytic virotherapy was first noted by George Dock in 1904. Similar to Coley, Dock noticed that a patient with acute leukemia underwent remission after acquiring an influenza infection [147]. Many other case reports followed over the years, eventually leading to hundreds of clinical trials testing oncolytic viruses [148]. In 2015, the first oncolytic viral therapy, talimogene laherparepvec (“T-VEC”), was approved by the FDA for use in metastatic melanoma [143,149]. T-VEC is an attenuated herpes simplex virus harboring various genetic deletions and insertions designed to enhance the antitumor immune response, such as the deletion of an immune-evasive viral gene *ICP47* and the insertion of a human GM-CSF gene [145]. Compared to GM-CSF administration alone, T-VEC led to a 4.4 month increase in median survival in a phase III trial in patients with advanced and metastatic melanoma [143,149].

## 5. Adoptive Cell Therapy

### 5.1. Tumor-Infiltrating Lymphocytes and Engineered T-Cell Receptors

The antitumor activity of T lymphocytes was first elucidated in the 1950s and 1960s with seminal discoveries made in allograft rejection of sarcomas in experimental rodent models [150,151]. In 1953, Mitchison investigated the passive transfer of tumor immunity via transplantation of lymph nodes from mice inoculated with lymphosarcomas to equivalently challenged, but non-inoculated, mice [152]. A decade later, two groups made similar observations of transferrable tumor immunity by isolating and transplanting the cells of lymphatic tissues, rather than the organs themselves. Cells were collected from the lymphatic ducts and spleens of donor animals previously immunized with sarcoma cells that developed palpable tumors. Administration of those lymphoid cells back into syngeneic and non-syngeneic recipients inoculated with tumors, saw sustained regression indicating that these lymphocytes were “educated” by prior exposure to tumor antigens [153,154].

The means to exploit these T lymphocytes for their antitumor potential was limited by the inherent difficulty of expanding cells ex vivo. In 1976, a group at the NIH first described the co-culture of isolated human bone marrow with conditioned media containing IL-2 that could induce growth and differentiation of bone marrow cells to T lymphocyte precursors [155]. With the advent of commercially synthesized IL-2, T lymphocytes could now be cultured in large quantities, or in the context of an adjuvant, to boost the therapeutic effects of tumor-sensitized and adoptively transferred T lymphocytes [156]. This subset of cytokine-activated lymphocytes was identified to be among those infiltrating the stroma of transplanted tumors. Dr. Steven Rosenberg’s group confirmed that tumor-infiltrating lymphocytes (TILs) isolated from resected tumor could recognize syngeneic tumor cells in vitro [157], as well as mediate durable antitumor responses when re-administered back into autologous animal models [158] and cancer patients with metastatic disease [159].

The relatively pure populations of CD8^+^ and CD4^+^ T cells cultured from resected tumors appeared to dissipate quickly when returned to patients, meaning that therapeutic responses were often transient. However, in 2002, prior application of a lymphodepleting, nonmyeloablative chemotherapy regimen, originally designed for allogeneic bone marrow transplantation [160], greatly enhanced TIL engraftment and clonal persistence in patients, with some individuals harboring up to 80% CD8^+^ T cells many months post-infusion [161,162]. Shortly thereafter, another milestone was achieved when the Rosenberg group retrovirally transduced patient-derived peripheral blood lymphocytes with a TCR recognizing the common melanoma antigen, MART-1. Objective cancer regression was observed in 2 out of 15 patients (13%) when engineered T lymphocytes were adoptively transferred back into patients [163], with a subsequent report demonstrating an improved 30% objective response rate [164]. Additional trials employing engineered TCRs targeting NY-ESO-1 in synovial cell sarcoma [165], the GD2 disialoganglioside in neuroblastoma [166], and carcinoembryonic antigen (CEA) in colorectal cancer [167] demonstrated objective clinical responses, thereby broadening the application to additional tumor types. More recently, personalized strategies using whole-exome sequencing of patient tumors has given researchers the ability to target neoantigens with high specificity [168,169]. Classically unmanipulated TIL therapy will continue to serve a patient population with shared and broadly targetable antigens [170], while more nuanced TILs recognizing neoantigens will continue to pace evolving therapies in the age of personalized medicine [171].

### 5.2. Chimeric Antigen Receptor T Cell (CAR-T Cell) Therapy

Although adoptive transfer of tumor-sensitized and antigen-reactive TILs with prefatory lymphodepletion and IL-2 dosing regimens had proven effective in the clinic, patient responses were often transient: shrinkage in metastatic lesions could occur, without objective response to treatment [172]. Native TCRs are often limited by their ability to recognize post-translationally or aberrantly modified proteins, such as those observed in tumor-associated antigens of malignant cells [173,174]. Likewise, T cells dependent on antigen presentation by MHC molecules are routinely hindered by MHC class I downregulation, a selective mechanism of tumor immune escape [175].

In 1989, an Israeli group at the Weizmann Institute of Science devised the first proof-of-concept strategy using an engineered chimeric antigen receptor (CAR) to circumvent the need for MHC-mediated antigen presentation for T-cell activation. By combining the intracellular T-cell receptor circuitry with the antigen-recognizing variable domains of an antibody raised against 2,4,6-trinitrophenyl (TNP), the researchers were able to elicit a non-MHC-restricted response in transfected T lymphocytes to target cells bearing TNP on their surface [176]. The unprecedented antibody-type specificity, now liberated from MHC presentation and paired to effector T-cell function, could conceivably target post-translationally modified proteins characteristic of tumor cells undergoing selection or escape. Several years later, the same group successfully generated CARs directed towards HER2, a cell surface antigen commonly overexpressed in adenocarcinomas. These CAR-T cells selectively lysed HER2^+^ cancer cells in vitro, providing evidence that mAbs directed towards common tumor cell antigens, could be reassembled into single chain variable fragments (scFvs) to facilitate immune effector function directly [177]. That same year, a joint venture between Weizmann Institute and the NIH expanded the spectrum of available targets by targeting folate receptors commonly overexpressed in ovarian carcinoma cells and further demonstrating the potential of adoptively transferring these CAR-T cells into cancer patients [178].

In an effort to improve CAR-T cell activation, CD28 costimulatory molecules were added in a single tandem gene product with the intracellular CD3ζ-chain. Tumor cells often lack costimulatory molecules entirely, a barrier to persistent activation in first-generation CAR-T cells. Much like conventional T cells, an “exhausted” phenotype was observed in T cells expressing first-generation constructs encountering tumor cells in excess. In contrast, second-generation CAR-T cells containing additional built-in CD28 costimulatory moieties demonstrated superior signaling functionality, persistence, and cytokine production [179,180], as well as antitumor activity [181]. Over the next decade, second-generation CARs would be the basis for many first-in-human studies: first targeting carbonic anhydrase IX (CAIX), an antigen commonly overexpressed in renal cell carcinoma (RCC), and shortly thereafter, the ovarian cancer–associated antigen α-folate receptor (FR). CAIX-directed CAR-T cells produced grade 2–4 liver toxicity in patients due to CAIX self-antigen present in normal bile duct epithelium, with no overall response in RCC tumors [182]. Likewise, FR-directed CAR-T cells in a parallel phase 1 study, produced no reduction in ovarian tumor burden, albeit with lower grade 1–2 toxicity and no detectable off-tumor or off-target responses [183].

Major clinical breakthroughs were not seen until several years later when CAR-T cell therapy strategy switched from targeting primarily solid tumors, to liquid tumors, such as B-cell lymphomas and leukemias. In 2008, a group at University of Washington pioneered a proof-of-concept clinical trial in which refractory B-cell lymphoma patients received 20 infusions of autologous CD20-directed CAR-T cells. Treatment was well-tolerated in patients, with minimal toxicities and modified T cells persisting up to 9 weeks. Of the seven patients, two were noted as having complete response to treatment [184]. With phase 1 clinical trials rapidly taking shape around B-cell targets, methods for manufacturing and validating clinical-grade autologous CAR-T cells were developed to support increasing demand [185]. Shortly thereafter, the Rosenberg group within the NCI’s Surgery Branch, demonstrated in vivo antigen-specific activity of CAR-T cells directed towards the B-cell-specific antigen, CD19, in advanced-stage follicular lymphoma [186]. Paralleling this seminal study, Dr. Carl June’s group at the University of Pennsylvania demonstrated specific and effective on-target killing of CD19^+^ malignant B cells in patients with advanced chronic lymphocytic leukemia (CLL) using CD19-directed CAR-T cells. In that study, two out of three patients experienced complete remission of disease, with a portion of CAR-T cells retaining potent effector function six months after initial infusion, indicating a possible memory CAR-T cell phenotype [187,188]. The CAR employed by the University of Pennsylvania possessed a 4-1BB (CD137) costimulatory domain, rather than CD28, that promoted in vivo persistence and antileukemic function that outperformed conventional CARs with either CD3ζ and CD28 costimulatory molecules or CD3ζ alone [187,189].

Over the next few years, both groups continued to advance the field by targeting various CD19^+^ hematological malignancies with great success. However, unanticipated and oftentimes severe neurological toxicities in the form of cytokine release syndrome were observed in patients. This toxicity can manifest as fevers, headaches, aphasia, and in some cases, hallucinations, delirium, and seizures [190]. Cytokine blockade strategies to control the abundance of systemically released cytokines, namely administering the IL-6-blocking antibody tocilizumab with and without corticosteroids, were developed to combat acute neural toxicity [191,192,193,194,195]. Although major clinical gains have been achieved with CD19^+^ hematologic malignancies, the same successes have yet to be fully realized in solid tumors. Despite abundant antigenic heterogeneity, difficulties in trafficking to tumor sites, and an intrinsic immunosuppressive tumor microenvironment [196,197], CAR-T cell therapies against solid tumor malignancies have entered early-phase clinical trials with varying degrees of success [Table 1].

## 6. Conclusions

The late 19th century observation that tumors could be treated with cocktails of heat-killed bacteria has proven highly influential. Unbeknownst to William Coley and his contemporaries, this would prove to be one of the first documented cases of tumor regression by induced activation of the immune system. Coley’s legacy would help spur subsequent hypotheses of immunosurveillance mechanisms capable of mediating steady-state tumor recognition and elimination. Over the next century, exploitation of these mechanisms was to become a major priority as immunotherapies continued to evolve.

Treatment regimens using recombinant cytokines that activate immune cell proliferation and effector functions are efficacious in treating selected patient populations. Likewise, strategies employing immune checkpoint blockade against tumor cells that express co-inhibitory molecules have reached clinical milestones once thought to be unachievable. Vaccines against tumor-associated antigens have demonstrated clinical benefit, with applications turning towards neoantigens as patient-specific tumor sequencing becomes feasible. FDA approval of the oncolytic virotherapy, T-VEC, may also offer another option for locally-acting immune stimulation and antitumor activity when resection, chemotherapy, or radiation are not amenable. Moreover, the success of CAR-T cell therapy in patients with hematological malignancies has established adoptive cell therapy as a viable treatment modality. As the costs associated with patient and tumor genome sequencing continue to decrease, the rapidly evolving “omics”-level of data acquisition and processing may enable precise treatment strategies for these patients. Deconvoluting patient-specific tumor heterogeneity with the assistance of “big data” may enable clinicians and researchers to select the best candidate immunotherapy from the start, while taking proactive approaches to overcome resistance mechanisms in an adaptive tumor microenvironment [198]. The previous ~150 years of immuno-oncology research without significant clinical success has now enabled “hockey stick” growth in exploration of the safety and efficacy of immune-centric therapies in clinical trials. Just as William Coley’s fundamental discoveries have shaped modern cancer immunotherapy, so too shall current efforts influence the future of cancer treatment.

## Figures and Tables

**Figure 1 toxins-12-00241-f001:**
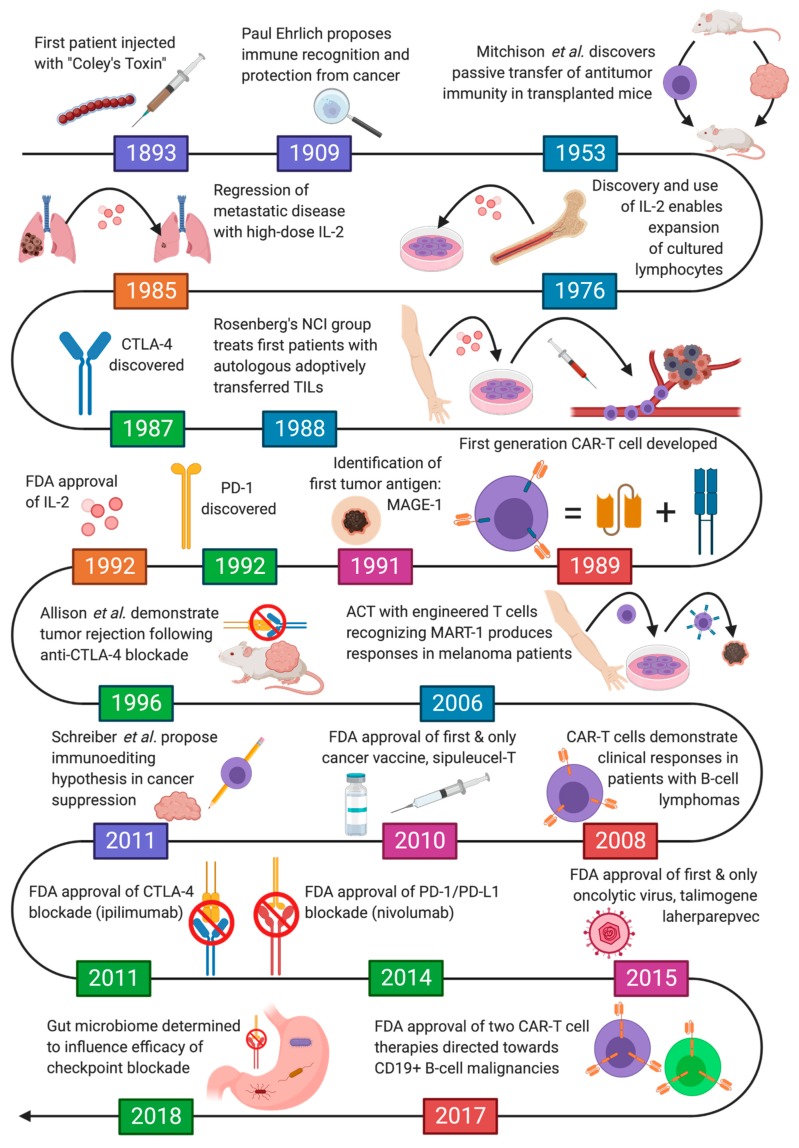
Milestones in the History of Cancer Immunotherapy.

**Table 1 toxins-12-00241-t001:** Summary of active clinical trials for CAR-T cell therapy in solid tumors.

Antigen Target	Cancer Type	Phase	ClinicalTrials.gov Designation
CD117	Osteoid Sarcoma, Ewing Sarcoma	I/II	NCT03356782
CD133	Liver, Pancreatic, Brain, Breas, Ovarian, Colorectal, Acute Myeloid and Lymphoid Leukemias	I/II	NCT02541370
	Osteoid Sarcoma, Ewing Sarcoma	I/II	NCT03356782
CD171	Neuroblastoma, Ganglioneuroblastoma,	I	NCT02311621
CEA	Colorectal	I/II	NCT02959151
	Lung, Colorectal, Gastric, Breast, Pancreatic	I	NCT02349724
EGFR	Colorectal	I/II	NCT03152435
EGFRvIII	Glioma, Glioblastoma, Gliosarcoma	I/II	NCT01454596
EpCAM	Colon, Esophageal, Pancreatic, Prostate, Gastric, Hepatic	I/II	NCT03013712
EphA2	Glioma	I/II	NCT02575261
ErbB	Head and Neck	I/II	NCT01818323
FRα	Urothelial Bladder	I/II	NCT03185468
GD2	Glioma	I/II	NCT03252171
	Neuroblastoma	I/II	NCT03373097
		I	NCT01822652
		II	NCT02765243
	Cervical	I/II	NCT03356795
	Osteoid Sarcoma, Ewing Sarcoma	I/II	NCT03356782
	Sarcoma, Osteosarcoma, Neuroblastoma, Melanoma	I	NCT02107963
GPC3	Hepatocellular Carcinoma	I/II	NCT02723942
			NCT02959151
			NCT03084380
HER2	Breast	I/II	NCT02547961
	Sarcoma	I	NCT00902044
IL-13Rα2	Glioma, Glioblastoma	I	NCT02208362
Mesothelin	Pancreatic	I/II	NCT02959151
	Cervical	I/II	NCT03356795
	Advanced Solid Tumors	I/II	NCT03615313
	Pancreatic, Ovarian, Mesothelioma	I	NCT02159716
	Malignant Pleural Disease, Mesothelioma, Lung, Breast	I	NCT02414269
MUC1	Cervical	I/II	NCT03356795
	Esophageal	I/II	NCT03706326
	Non-Small Cell Lung	I/II	NCT03525782
	Osteoid Sarcoma, Ewing Sarcoma	I/II	NCT03356782
	Intrahepatic Cholangiocarcinoma	I/II	NCT03633773
MUC-16	Ovarian	I	NCT02498912
PSMA	Urothelial Bladder	I/II	NCT03185468
	Cervical	I/II	NCT03356795
	Prostate	I	NCT01140373

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
