# Peer review of "Talkin’ Toxins: From Coley’s to Modern Cancer Immunotherapy"

_toxins, 2020, doi:10.3390/toxins12040241_

Round 1
Reviewer 1 Report
The review entitled: "Talkin’ Toxins: From Coley’s to Modern Cancer Immunotherapy" is a well written, well presented and a well organised review, pleasant to read. The review summarises important discoveries that have led to the most current anti-cancer immunotherapies. The authors may however wish to add to the sentence written in line 106 that the clinical evidence observed in human with compromised immune systems were all related to virus-induced cancer and that when the immune system is compromised the majority of cancers observed are blood-related cancer snot solid cancer.
Line 187, could the authors give more details of what is meant by "proven efficacious"?
Line 244 the authors may wish to add the name of the anti-PD-L1 antibody (I assume Atezolizumab) for completeness.
Author Response
The authors may however wish to add to the sentence written in line 116 that the clinical evidence observed in human with compromised immune systems were all related to virus-induced cancer and that when the immune system is compromised the majority of cancers observed are blood-related cancer snot solid cancer.
The sentences following line 116 indicate that those cancers were primarily virally-induced and liquid cancers. We also added to line 121 that Kaposi Sarcomas are virus-induced.
Line 200, could the authors give more details of what is meant by "proven efficacious"?
Clarified that efficacy was observed in the two cited publications (66 & 67) where treatment strategies achieved enhanced overall survival endpoints.
Lines 257-259 the authors may wish to add the name of the anti-PD-L1 antibody (I assume Atezolizumab) for completeness.
We added a sentence to clarify the name of anti-PD-L1 antibody (atezolizumab) and added an explanation/indications for atezolizumab FDA approval.
Reviewer 2 Report
Dear Authors,
This review is very comprehensive and very well written. It has been a priviledge to read it.
Abstract
Line 2: Please clarify the word "appreciated" or replace with "studied"
Line 7: dating "back" over...
Line 9: had been "dotted" with ...
Introduction
Line 20-23: Please make a shorter sentence
Figure 1: First report by Coler et al. was in 1893.
- Please also mention observations made in early 19s of oncolytic virotherapy: Bierman 1953, Dock 1904, De Pace 1912.
- Please mention: Couzin-Frankel J. Breakthrough of the year 2013. Cancer immunotherapy. Science. 2013;342(6165):1432–1433
- line 2018: correct "efficacy"
Line 39: "episode" instead of "encounter"
Line 47-48: please provide details on the side effects of the S.Pyogenes treatment
Line 53-54: Please provide information about the safety and tolerability of Coley's treatment as described in the 150 papers.
Line 59-60: please provide the FDA reasons for re-categorizing the Coley's toxin
Lines 46.50 and 54: please provide dates
Lines 90-92: provide publications to support this final
Line 128: "cure their disease"
Line 140: name of the therapies?
Line 143: the "therapeutic" immonomodulatory...
Line 157-159: Please provide information on the side effects since IL-2 induces acute vasoconstriction (Rasetti, JAP, 1995)
Line 202-204: unclear
Line 220: please discuss why the effects are more modest and compare with benefit of alternative therapies
Line 249: please define the non-responders types and their percentage?
Line 260-267: Please clarify at what stage and why vaccines were introduced in addition to the immunotherapy?
Line 277-283: In which cancer types?
Line 290: defiition of "peptide-pulsed"?
Line 293-295: Please add the historical perspective about the use of viruses (oncolytic virotherapy in early 19s)
Line 303: Please jsutify the short survival and compare with alternative therapies.
Line 314-315: Please provide more details about efficacy if possible
Line 368: Definition of CEA?
Line 373: Please explain how this personalisation will be achieved?
Line 378: Please explain how transient?
Line 419: details about "liquid tumors"
Line 434: Provide details of the cytokines syndrome side effects and if this could be treated with corticoids
Conclusion:
The conclusion is clear but please provide more historical perspective regarding the virotherapy and also more details on the side effects. Please add a statement on the progress still to be made with personailsed therapies.
Author Response
Abstract
Line 11: Please clarify the word "appreciated" or replace with "studied"
Changed to “studied”
Line 13: dating "back" over...
Changed to “back”
Line 15: had been "dotted" with ...
Changed to “dotted”
Introduction
Line 26-29: Please make a shorter sentence
Sentence split in two. Second sentence adjusted for clarity.
Figure 1: First report by Coler et al. was in 1893.
The figure was adjusted to reflect date of published report, rather than initial injection of Coley’s Toxin.
Please also mention observations made in early 19s of oncolytic virotherapy: Bierman 1953, Dock 1904, De Pace 1912.
We added oncolytic virus milestone to Figure 1 and added an entire subsection on oncolytic virotherapy (new section 4.4). Thank you for this suggestion!
Please mention: Couzin-Frankel J. Breakthrough of the year 2013. Cancer immunotherapy. Science. 2013;342(6165):1432–1433
We were unable to fit this in the figure; however, we now reference this publication in the text (line 25) reflecting its focus on the significant recent leaps in cancer immunotherapy.
Figure 1, “line 2018”: correct "efficacy"
Corrected. Thanks for catching that!
Line 44: "episode" instead of "encounter"
Corrected
Line 52-53: please provide details on the side effects of the S.Pyogenes treatment
Added additional details about the safety profile/side effects of Coley’s Toxins.
Line 54-55: Please provide information about the safety and tolerability of Coley's treatment as described in the 150 papers.
Added additional details about the safety profile/side effects of Coley’s Toxins.
Line 68-69: please provide the FDA reasons for re-categorizing the Coley's toxin
Provided and clarified the implications
Lines 51 and 54: please provide dates
Dates have been added.
Lines 102: provide publications to support this final
Citations were added.
Line 138: "cure their disease"
Corrected.
Lines 150-151: name of the therapies?
We revised the sentence to indicate that this section will describe those “current therapies” in detail.
Line 154: the "therapeutic" immonomodulatory...
Revised as suggested.
Line 167-168: Please provide information on the side effects since IL-2 induces acute vasoconstriction (Rasetti, JAP, 1995)
We inserted a sentence and citation about prominent IL-2 treatment side effects and measures taken to overcome these treatment related toxicities.
Lines 214-216: unclear
Revised for clarity.
Line 220: please discuss why the effects are more modest and compare with benefit of alternative therapies
Many of these clinical trials and accompanying retrospective studies have yet to reveal precisely why patients respond to checkpoint inhibitors disproportionately. To the end of this section (lines 263-267) we added a description of innate/adaptive resistance mechanisms that may prevent uniformity in patient response.
Line 263: please define the non-responders types and their percentage?
We have added an explanation of patient non-responder subsets and clarified efforts to combat checkpoint inhibitor therapy resistance (lines 263-267). Exact percentages of patient non-responders varies by tumor type and stage, and is therefore much too comprehensive for this review.
Line 260-267: Please clarify at what stage and why vaccines were introduced in addition to the immunotherapy?
Vaccines have been explored throughout the decades of cancer immunology discovery and cancer immunotherapy development. Indeed, studies by Prehn and Main supporting immunosurveillance essentially used tumor cell vaccines to induce immunity to subsequent tumor challenged (described in lines 92-102).
Line 294-300: In which cancer types?
This is beyond the scope of the sentence. Its purpose is to broadly introduce different categories of tumor-associated antigens rather than give a list of all the dozens of types of cancer each antigen has been identified in.
Line 307: definition of "peptide-pulsed"?
We have added a definition.
Line 311-313: Please add the historical perspective about the use of viruses (oncolytic virotherapy in early 19s)
We have added an entire subsection on oncolytic virotherapy (lines 351-373; section 4.4)
Line 321: Please justify the short survival and compare with alternative therapies.
We think this is speculative to compare survival benefits outside of a trial.
Line 314-315: Please provide more details about efficacy if possible
No neoantigen vaccines have been tested in phase III trials, making it difficult to make claims about efficacy
Line 409: Definition of CEA?
The definition was added (carcinoembryonic antigen)
Line 415: Please explain how this personalisation will be achieved?
Personalization using “whole-exome sequencing of patient tumors” to identify neoantigens for TIL-targeting is explained in the prior sentence (lines 411-412).
Line 419: Please explain how transient?
We clarified what was meant by “transient” patient responses as a potential therapy drawback.
Line 460: details about "liquid tumors"
Changed to “B-cell” to clarify CD20 and CD19-directed therapies described in previous sentence and latter sentence with respect to “liquid” or hematologic B-cell malignancies.
Line 475: Provide details of the cytokines syndrome side effects and if this could be treated with corticoids
We have added details about cytokine release syndrome side-effects with appropriate citations and details about treating side-effects with/without corticosteroids, since steroid treatment is not always appropriate.
Conclusion: The conclusion is clear but please provide more historical perspective regarding the virotherapy and also more details on the side effects. Please add a statement on the progress still to be made with personailsed therapies.
We have included a summary statement and perspective for oncolytic virotherapy; a statement about progress and future perspectives for personalized therapies incorporating “omics”-level data in designing treatment strategies; and included a citation about how “big data” can enable clinicians/researches to make treatment decisions.
This manuscript is a resubmission of an earlier submission. The following is a list of the peer review reports and author responses from that submission.